# High-quality whole-genome sequence analysis of *Lactobacillus paragasseri* UBLG-36 reveals oxalate-degrading potential of the strain

Yogita Mehra, Pragasam Viswanathan[ID]*

Renal Research Lab, Centre for Bio-Medical Research, School of Bio-Sciences and Technology, Vellore Institute of Technology, Vellore, Tamil Nadu, India

* pragasam.v@vit.ac.in

**Data Availability Statement:** All relevant data are within the paper and its Supporting Information files.

## Abstract

*Lactobacillus paragasseri* was identified as a novel sister taxon of *L. gasseri* in 2018. Since the reclassification of *L. paragasseri*, there has been hardly any report describing the probiotic properties of this species. In this study, an *L. paragasseri* strain UBLG-36 was sequenced and analyzed to determine the molecular basis that may confer the bacteria with probiotic potential. UBLG-36 was previously documented as an *L. gasseri* strain. Average nucleotide identity and phylogenomic analysis allowed accurate taxonomic identification of UBLG-36 as an *L. paragasseri* strain. Analysis of the draft genome (~1.94 Mb) showed that UBLG-36 contains 5 contigs with an average G+C content of 34.85%. Genes essential for the biosynthesis of bacteriocins, adhesion to host epithelium, stress resistance, host immunomodulation, defense, and carbohydrate metabolism were identified in the genome. Interestingly, *L. paragasseri* UBLG-36 also harbored genes that code for enzymes involved in oxalate catabolism, such as formyl coenzyme A transferase (*frc*) and oxalyl coenzyme A decarboxylase (*oxc*). *In vitro* oxalate degradation assay showed that UBLG-36 is highly effective in degrading oxalate (averaging more than 45% degradation), a feature that has not been reported before. As a recently identified bacterium, there are limited genomic reports on *L. paragasseri*, and our draft genome sequence analysis is the first to describe and emphasize the probiotic potential and oxalate degrading ability of this species. With results supporting the probiotic functionalities and oxalate catabolism of UBLG-36, we propose that this strain is likely to have immense biotechnological applications upon appropriate characterization.

## Introduction

The presence of oxalate degrading bacteria in the human gut microbiome is essential as humans lack the enzymes required to metabolize endogenous and dietary oxalate. Free oxalate in the gut is mostly eliminated via urine or feces (as insoluble calcium oxalate conjugate) or metabolized by oxalate degrading gut bacteria. However, disruption of oxalate homeostasis

**Funding:** - Pragasam Viswanathan - Indian Council of Medical Research (No.5/9/1094/2013-NUT)

**Competing interests:** The authors have declared that no competing interests exist.

leads to over-accumulation of oxalate, which is a notable cause of several pathological conditions such as hyperoxaluria [1], calcium oxalate urolithiasis [2], cardiomyopathy [3], and renal failure [4]. Consequently, in recent years, there has been an increased focus on the use of probiotics for the prevention of oxalate related disorders [5], mostly those that belong to the genera *Lactobacillus* and *Bifidobacterium* [6], perhaps because they are considered generally recognized as safe (GRAS) for human consumption. Among the *Lactobacillus* species, strains of *L. acidophilus* and *L. gasseri* have been most widely studied for their general ability to utilize oxalate and other substrates such as glucose as a source of energy.

*L. gasseri* is an obligate homofermentative, facultative anaerobe in the family of lactic acid bacteria (LAB) first identified in 1980 [7]. Since the availability of the first complete genome sequence of *L. gasseri*, several draft-phase or complete genomes of this organism have been deposited in public databases. In a recent study by Tanizawa and colleagues [8], a distinct species of *L. gasseri* was identified based on whole-genome average nucleotide identity (ANI) analysis. This new species of bacteria was termed *L. paragasseri*, and it represents a novel sister taxon of *L. gasseri*. A study comparing the genetic diversity of 13 strains of *L. gasseri* and 79 strains of *L. paragasseri* showed a clear distinction between these two species, particularly in their clustered regularly interspaced short palindromic repeats (CRISPR), bacteriocins, and their carbohydrate utilization patterns [9]. However, despite the genetic differences, the comparative pan-genome analysis also indicates a higher rate of interspecies gene exchange between *L. gasseri* and *L. paragasseri* [9]. Like its sister taxon, *L. paragasseri* harbors genes that code for enzymes involved in oxalate catabolism. In this two-step process, formyl coenzyme A transferase (EC 2.8.3.16) encoded by the *frc* gene catalyzes the transfer of CoA from formyl-CoA to oxalate [10]. This activated oxalate molecule is then decarboxylated by the enzyme oxalyl coenzyme A decarboxylase (EC 4.1.1.8) encoded by the *oxc* gene [10]. While the oxalate utilizing abilities of *L. gasseri* and its probiotic relevance has been studied [11, 12], there is hardly any evidence on *L. paragasseri* for the same.

In this study, UBLG-36, a commercially available strain hitherto identified as *L. gasseri*, was sequenced and re-classified as *L. paragasseri* based on whole-genome relatedness. We performed several bioinformatic analyses to better understand its genomic features, specifically those that may likely contribute to the probiotic properties of the strain. We also identified genes putatively associated with oxalate catabolism and analyzed the ability of the strain to degrade oxalate *in vitro*.

## Materials and methods

### Bacterial source and identification

A commercial probiotic strain UBLG-36, initially identified as *L. gasseri*, was purchased from Unique Biotech Limited (Hyderabad, India). *L. acidophilus* DSM 20079 and *L. casei* ATCC 334 were purchased from the German Collection of Microorganisms and Cell Cultures GmbH (Germany) and American Type Culture Collection (USA), respectively. Strains were cultured in *Lactobacillus* De Man, Rogosa, and Sharpe (MRS) selective medium at 37˚C under the aerobic condition for 18–24 hours. UBLG-36 was subjected to 16S rRNA sequencing, and verification of its accurate taxonomic identification was performed on the EzBiocloud (https://www.ezbiocloud.net/) online platform.

### Genome sequencing, assembly, and annotation

Unless otherwise mentioned, only the default parameters were used in the software. Whole-genome sequencing of the strain UBLG-36 was performed on Illumina Miseq platform by Illume Gene India LLP Company (Bengaluru, India), and the raw sequencing reads were

uploaded to Galaxy Europe genome analysis tool (http://usegalaxy.eu/). De novo assembly of the trimmed reads was performed on Unicycler 0.4.8.0 [13] using the normal bridging mode. Contigs shorter than 100 base pairs were excluded, and the assembly quality was improved using the Multi-Draft-based Scaffolder (MeDuSa) [14]. Annotation of the draft assembly was performed by the NCBI Prokaryotic Genome annotation pipeline (PGAP) [15]. The assembled genome was also analyzed using the RASTtk pipeline of the Rapid Annotations using Subsystems Technology (RAST) online platform (http//rast.nmpdr.org) [16]. EggNOG mapper 4.5 [17] was used to determine the clusters of orthologous groups (COGs) of proteins. Carbohydrate-active enzymes (CAZymes) were searched against the CAZy database using the dbCAN2 meta server [18]. Secondary metabolite-related genes/gene clusters were predicted using antiSMASH 3.0 [19]. ResFinder tool [20] and the Comprehensive Antibiotic Resistance Database (CARD) [21] was used to test for antibiotic resistance genes. Prophage genes were searched using PHASTER [22], and the prediction of the CRISPR-Cas system was carried out using CRISPRCas Finder [23]. Protein sequences that were likely to determine the putative probiotic properties of UBLG-36 were searched individually against the NCBI Conserved Domain Database [24].

## Phylogenomic analysis

Representative complete genome sequences of *L. gasseri* and *L. paragasseri* were obtained from the NCBI database (S1 Table). 16S rDNA sequences were mined from these complete representative genomes, and together with the 16S rDNA sequence of UBLG-36, pairwise sequence similarities were calculated via the Type (Strain) Genome Server (TYGS) [25] available at https://tygs.dsmz.de/. Phylogenies based on maximum likelihood and maximum parsimony were inferred by the TYGS web server. Draft genome sequence of UBLG-36 and eleven representative complete genome data were uploaded to the JspeciesWS server [26] and the TYGS [25] for calculating ANI values and pairwise comparison of genome sequences, respectively.

Coding sequences (CDS) of *oxc* and *frc* from the UBLG-36 genome and the eleven representative complete genomes of *L. gasseri* and *L. paragasseri* were mined from the NCBI database. These CDS were aligned using the MAFFT webtool [27], and the residue-wise confidence scores were obtained using the GUIDANCE2 [28] application. Phylogenetic tree construction was performed by the MAFFT tool using the Neighbor-Joining method with 1000 bootstrap resampling. Default software parameters were used for all phylogenomic analyses.

## Oxalate degradation assay

The oxalate degrading ability of UBLG-36 was determined *in vitro*. Two MRS culture media with 10 mM and 20 mM sodium oxalate (MRS-ox) (pH 5.5) were prepared as described earlier [29]. Culture broths were inoculated with 1% bacterial culture grown overnight and incubated at 37°C for 1, 5 and 10 days under aerobic conditions. MRS-ox broth without bacterial inoculum was used as a control. *L. acidophillus* DSM 20079 (NZ_CP020620; harboring both *oxc* and *frc*) was used as the positive control, while *L. casei* ATCC 334 (NC_008526; lacking *oxc* and *frc*) served as the negative control. Before the analysis, culture broth and control media were centrifuged at 4000 g for 10 minutes, and the supernatant was filtered through a 0.45-micron filter. Oxalate concentration in the filtrate was measured in triplicates using an oxalate assay kit (Sigma, USA) as per the manufacturer's instructions.

## Statistical analysis

The results are presented as mean ± standard error of mean (SEM). A one-way analysis of variance (ANOVA) followed by Tukey's test was performed to compare the concentration of oxalate degraded by the bacterial strains. A two-way analysis of variance (ANOVA) followed by

Tukey's test was performed to compare percentage oxalate degradation at 10 mM and 20 mM MRS-Ox at three different time points by the bacterial strains. Comparison between datasets were considered statistically significant at $p<0.05$ (indicated as $^{***}p<0.001$, $^{\#\#\#}p<0.001$, and $^{\#\#}p<0.01$). Statistical analysis was performed on Graph Pad Prism 9.2.0 for Windows, Graph-Pad Software, San Diego, California USA (www.graphpad.com).

## Results and discussion

### Initial identification

The commercially purchased *L. gasseri* UBLG-36 strain was subjected to an initial verification by 16S rRNA sequencing. Sequencing identified UBLG-36 as a strain belonging to *L. paragasseri* and not *L. gasseri*. To better understand the reason behind the misidentification of species assignment, we performed a detailed phylogenomic analysis (see below) after sequencing the draft genome.

### Phylogenomic analysis

To verify the result of the initial 16S rRNA sequencing, we compared the 16S rDNA sequence of UBLG-36 (obtained from the draft genome) to those of the reference genomes (comprising complete genome sequences of 4 *L. paragasseri* and 7 *L. gasseri* strains). Although UBLG-36 formed a separate clade with high similarity to four other *L. paragasseri* strains (JCM 5343, JV V03, NCK1347, and NCTC13720), four *L. gasseri* strains (EJL, MGYG-HGUT-02387, HL70, and HL75) were also found within this clade (Fig 1A). As whole-genome comparison can yield better microbial resolution and identification than 16S rDNA analysis, we performed a pair-wise comparison of the UBLG-36 draft genome with all eleven available complete genome sequences. Consistent with the results of the 16S rDNA comparison, UBLG-36 showed high whole-genome relatedness with three *L. gasseri* strains (EJL, MGYG-HGUT-02387, and HL70) than the *L. paragasseri* strains (Fig 1B). ANI calculation also showed that UBLG-36 shared more than 98% identity with *L. gasseri* EJL, *L. gasseri* MGYG-HGUT-02387, *L. gasseri* HL70, and *L. gasseri* HL75 (S2 Table). According to a previous report, several strains of *L. gasseri* available in the public databases are misidentified and are likely members of *L. paragasseri* species [30]. To investigate the likelihood of the four *L. gasseri* strains (EJL, MGYG-HGUT-02387, HL70 and HL75HL70 and HL75) belonging to *L. paragasseri* species, we calculated the ANI values of their genomes on the JspeciesWS server through pairwise comparison at the 95% threshold with all other publicly available complete genomes of *L. gasseri* and *L. paragasseri*. ANI values indicated that EJL and MGYG-HGUT-02387 shared more than 97% identity with *L. paragasseri* strains, whereas, with *L. gasseri* strains, the identity was less than 95%, which is less than the widely accepted threshold (Table 1). Although UBLG-36 was purchased as an *L. gasseri* strain, our analyses thus provide evidence for the taxonomic identification of UBLG-36 as *L. paragasseri*. Also, strains EJL, MGYG-HGUT-02387, HL70, and HL75, currently identified as *L. gasseri*, should be considered for reclassification into *L. paragasseri*.

### General genome features

The bacterial strain UBLG-36 (henceforth referred to as *L. paragasseri* UBLG-36), when subjected to whole-genome sequencing using Illumina MiSeq platform generated 4,313,094 paired-end reads (2 x 151 bp) sequences. The reads were uploaded to Galaxy Europe web genome analysis platform for quality assessment and read trimming. The resulting 3,513,720 paired-end reads were de novo assembled using Unicycler 0.4.8.0, and the assembly was further improved using MeDuSa. The final draft genome sequence of UBLG-36 showed 5 contigs

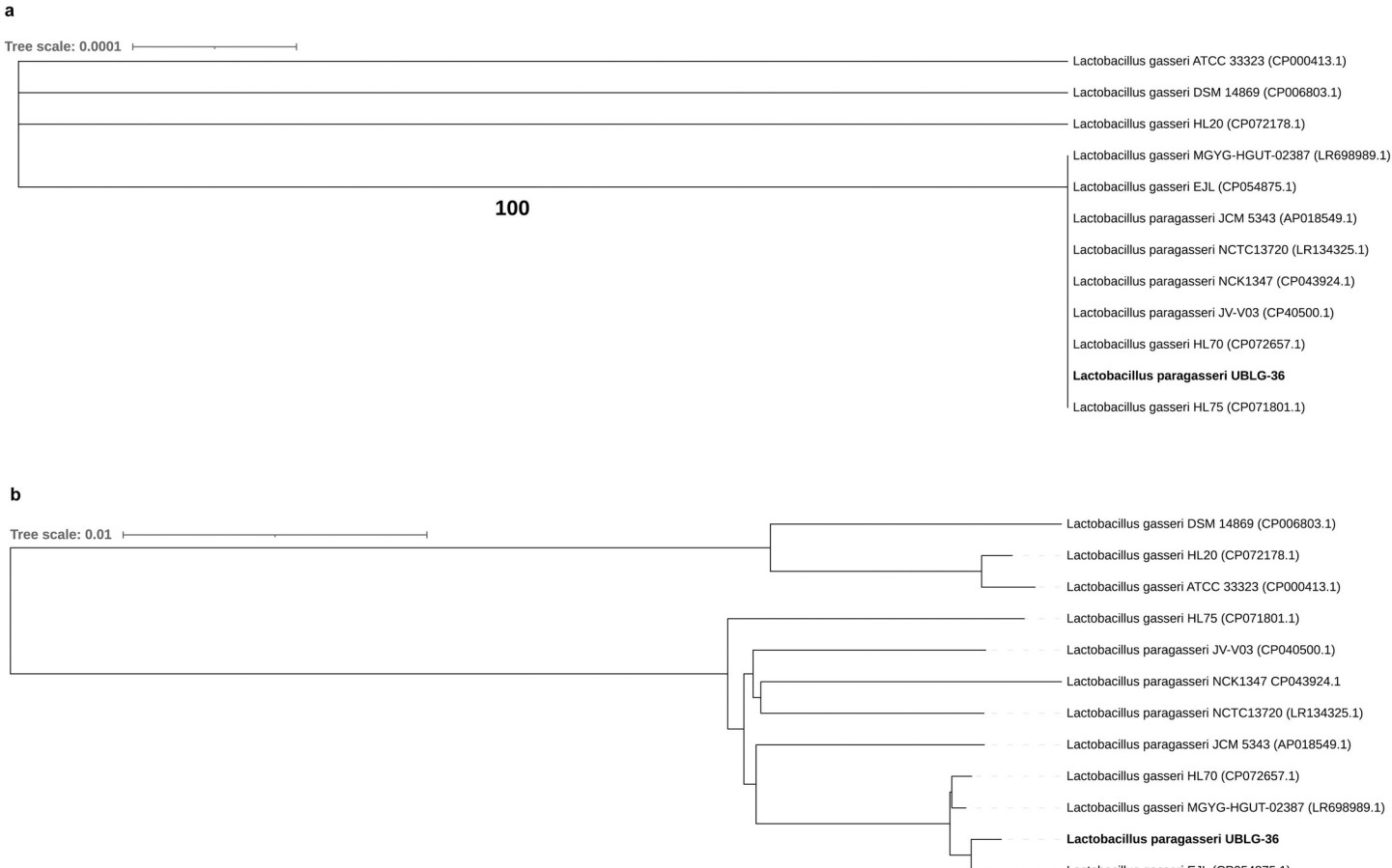

**Fig 1. Phylogenomic analysis of *Lactobacillus paragasseri* UBLG-36.** (a) Phylogenetic relationship between *Lactobacillus paragasseri* UBLG-36 and other *L. gasseri* and *L. paragasseri* strains. Phylogenetic tree construction was based on the pairwise sequence similarities of the 16S rDNA sequences mined from the complete representative genome sequences and draft genome sequence of UBLG-36. Tree calculation and inference were carried out on the TYGS web server (http://tygs.dsmz.de/). *L. paragasseri* UBLG-36 is depicted in bold. (b) Phylogenomic comparison of *Lactobacillus paragasseri* UBLG-36 with representative complete genomes of other *L. gasseri* and *L. paragasseri* strains. A pairwise comparison of the whole genomes was carried out on the TYGS web server, and the tree was inferred from GBDP distances calculated from the genome sequences. *L. paragasseri* UBLG-36 is depicted in bold.

covering a total length of 1,941,907 bp, with a G + C content of 34.85% and an $N$50 of 77,703 bp (Table 2). Annotation of the draft genome using the NCBI PGAP predicted a total of 1918

**Table 1. ANI values of *Lactobacillus gasseri* EJL, *L. gasseri* MGYG-HGUT-02387, *L. gasseri* HL70 and *L. gasseri* HL75 with other *L. gasseri* and *L. paragasseri* strains.**

| Genomes | EJL | | MGYG-HGUT-02387 | | HL70 | | HL75 | |
|---|---|---|---|---|---|---|---|---|
| | ANIb[a] [%] | Aligned [%] | ANIb[a] [%] | Aligned [%] | ANIb[a] [%] | Aligned [%] | ANIb[a] [%] | Aligned [%] |
| *Lactobacillus paragasseri* JCM 5343 | 98.39 | 78.97 | 98.39 | 80.67 | 98.30 | 82.60 | 97.86 | 84.45 |
| *Lactobacillus paragasseri* NCTC13720 | 97.86 | 77.83 | 98.10 | 79.48 | 97.94 | 83.14 | 97.81 | 85.22 |
| *Lactobacillus paragasseri* JV-V03 | 98.17 | 76.24 | 98.17 | 78.36 | 98.17 | 80.76 | 97.92 | 85.16 |
| *Lactobacillus paragasseri* strain NCK1347 | 97.97 | 75.72 | 97.98 | 77.89 | 97.93 | 80.30 | 97.00 | 85.78 |
| *Lactobacillus gasseri* ATCC 33323 | 93.26 | 72.49 | 93.11 | 74.68 | 93.27 | 76.90 | 92.89 | 82.40 |
| *Lactobacillus gasseri* DSM 14869 | 93.07 | 71.52 | 93.11 | 75.34 | 93.05 | 76.06 | 93.13 | 82.05 |
| *Lactobacillus gasseri* HL20 | 93.23 | 71.53 | 93.14 | 73.88 | 93.31 | 76.27 | 93.07 | 80.85 |

[a]ANIb refers to Average Nucleotide Identity Blast.

genes, including 1766 protein-coding genes, 3 rRNAs, 52 tRNAs, 3 ncRNAs, 1 tmRNA and 94 pseudogenes (Table 2). There were no plasmid sequences identified in the genome. Both ResFinder and CARD revealed that the genome of UBLG-36 harbored no antibiotic-resistant genes. Analysis of the genome on RAST using the annotated genes provided a general overview of the coded biological features with a subsystem coverage of 27% (Table 3).

## Functional classification

Of the predicted protein sequences queried in EggNOG mapper 4.5, 1636 proteins (92.64%) were assigned to 18 clusters of orthologous groups (COG) classes (Fig 2). Function unknown (S, 362 proteins) was the most common among all categories. Other proteins were classified into functional categories such as translation, ribosomal structure and biogenesis (J, 157 proteins); transcription (K, 146 proteins); replication, recombination and repair (L, 123 proteins); cell cycle control, cell division, chromosome partitioning (D, 31 proteins); defense mechanisms (V, 53 proteins); signal transduction mechanisms (T, 29 proteins); cell wall/membrane/envelope biogenesis (M, 93 proteins); Cell motility (N, 7 proteins); intracellular trafficking, secretion, and vesicular transport (U, 42 proteins); posttranslational modification, protein turnover, chaperones (O, 38 proteins); energy production and conversion (C, 54 proteins); carbohydrate transport and metabolism (G, 142 proteins); amino acid transport and metabolism (E, 92 proteins); nucleotide transport and metabolism (F, 87 proteins); coenzyme transport and metabolism (H, 39 proteins); lipid transport and metabolism (I, 36 proteins); inorganic ion transport and metabolism (P, 97 proteins); secondary metabolites biosynthesis, transport and catabolism (Q, 8 proteins).

## Carbohydrate-active enzymes (CAZymes)

Carbohydrate metabolism is gaining popularity as a trait supporting the probiotic potential of LAB [31]. Carbohydrate metabolism provides the main source of metabolic energy in LAB and plays an important role in the survival and fitness of *Lactobacillus* in their ecological niche by contributing to cellular processes such as energy production and stress response [32]. For example, in *L. paragasseri* presence of CAZymes have been shown to degrade non-digestible oligosaccharide, such as ketose and fructo-oligosaccharides thereby, selectively promoting growth and survival of the species within the host [33]. The genome of UBLG-36 was searched

**Table 2. General genome features of *Lactobacillus paragasseri* UBLG-36.**

| Attribute | Value |
|---|---|
| Genome size (bp) | 1,941,907 |
| Number of contigs | 5 |
| DNA G + C content (%) | 34.85 |
| *N50* (bp) | 77,703 |
| Total number of genes | 1,918 |
| Total number of protein-coding genes | 1,766 |
| rRNAs | 3 |
| tRNAs | 52 |
| ncRNAs | 3 |
| tmRNAs | 1 |
| Pseudogenes | 94 |
| Plasmid | 0 |
| Prophages (intact) | 1 |
| CRISPR arrays | 1 |

**Table 3. General overview of biological subsystem distribution of the annotated genes.**

| Description | Value | Percent (%) |
|---|---|---|
| Cofactors, vitamins, prosthetic groups, pigments | 46 | 6.9 |
| Cell wall and capsule | 48 | 7.2 |
| Virulence, disease and defense | 40 | 6 |
| Potassium metabolism | 9 | 1.3 |
| Miscellaneous | 3 | 0.4 |
| Phages, prophages, transposable elements, plasmids | 18 | 2.7 |
| Membrane transport | 25 | 3.7 |
| Iron acquisition and metabolism | 4 | 0.6 |
| RNA metabolism | 31 | 4.6 |
| Nucleosides and nucleotides | 67 | 10 |
| Protein metabolism | 123 | 18.4 |
| Cell division and Cell cycle | 4 | 0.6 |
| Regulation and cell signaling | 14 | 2.1 |
| Secondary metabolism | 2 | 0.3 |
| DNA metabolism | 57 | 8.5 |
| Fatty acids, lipids, and isoprenoids | 25 | 3.7 |
| Dormancy and sporulation | 5 | 0.7 |
| Respiration | 13 | 1.9 |
| Stress response | 6 | 0.9 |
| Metabolism of aromatic compounds | 2 | 0.3 |
| Amino acids and derivatives | 39 | 5.8 |
| Sulfur metabolism | 2 | 0.3 |
| Carbohydrates | 83 | 12.4 |

for CAZymes, and only those annotations that matched two or more tools in the dbCAN meta server were considered (S1 Fig). CAZymes analysis showed that UBLG-36 contains 29 genes that encode glycoside hydrolases (GHs), 26 genes that encode glycosyltransferases (GTs), 1 gene for carbohydrate esterase (CE), and 1 gene for carbohydrate-binding modules (CBMs) (S3 Table). These genes are essential for the bacteria's adaptation to the gastrointestinal environment and its interaction with the host since they are involved in the metabolism and assimilation of complex non-digestible carbohydrates [34]. For example, O-linked glycosylation on serines catalyzed by GTs can produce structures that are similar to mucins and may also facilitate adhesion to host cell mucoproteins [35]. Therefore, we believe that the presence of these enzymes will aid UBLG-36 in its survival, competitiveness, and persistence within the host.

## Secondary metabolites

Bacteriocins constitute a significant class of antimicrobial peptides produced by LAB [36]. These heat-labile, antimicrobial peptides have been used in the diary and cosmetic industry and in human and veterinary medicine for their ability to inhibit spoilage and pathogenic bacteria [37]. Prediction of secondary metabolites related genes or gene clusters using antiSMASH showed three biosynthetic clusters of bacteriocins with high overall similarity to Gassericin-T, Gassericin-S, and Acidocin-B (S4 Table).

## Prophages

Prophages are commonly found in genomes of *Lactobacillus* species [38]. Only one complete intact prophage locus (204,490 bp– 244,972 bp) was identified with a GC content of 36.41%

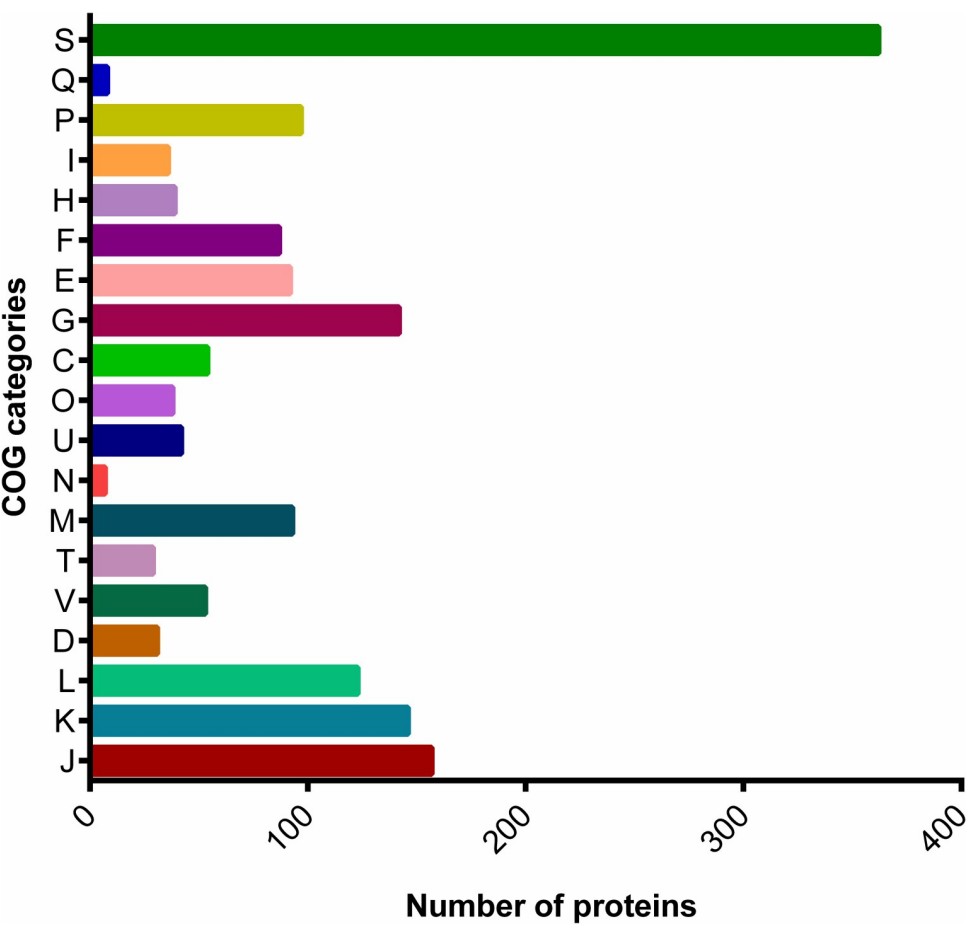

**Fig 2. COG functional categories assigned to the proteins of *Lactobacillus paragasseri* UBLG-36.**

(S5 Table). A 61 bp direct repeats (5′ –CGGAGCGTCATATCAGTGATTTGGGTGAGATTC–GAACTCACGACCCACGGTTTAG—AAGACC–3′ ) flanking the phage region was also identified. This represented the core regions of phage attachment sites (attL and attR). Although the exact role of phage elements in *Lactobacillus* is unknown, their presence may confer to the variation in the genome during the evolution of UBLG-36.

## CRISPR-Cas system

CRISPR refers to the short and highly conserved repeat regions in the genome that are interspaced with variable sequences (spacers) and are often located adjacent to CRISPR-associated (Cas) genes [39]. More than 40% of sequenced bacterial genomes show the presence of CRISPR, which provides immunity to the bacteria against foreign genetic elements [40]. The genome of UBLG-36 was analyzed for the presence of the CRISPR-Cas9 system using the CRISPRcas finder. UBLG-36 genome contains 1 CRISPR locus consisting of a 761-nucleotide region with 11 spacers (S6 Table). A Cas system of Type II-A (1,626,757 bp- 1,632,979 bp) was detected with four Cas genes; *cas1*, *cas2*, *cas9*, and *csn2*.

## Putative probiotic properties

The genome of UBLG-36 carries several protein-coding genes, which may determine its putative probiotic properties. Genes or proteins involved in acid and bile tolerance, adherence to

the intestinal mucosa, resistance to temperature changes, host immunomodulation, antimicrobial activity, and intrinsic defense were identified (Table 4). These features can provide selective survival advantages and are important in supporting the probiotic potential of the strain [41]. The genome of UBLG-36 contains genes such as *Nhac* that codes for Na$^+$-H$^+$ antiporter (MBO3730586) as well as genes for F0F1 ATPase synthase (MBO3730797-MBO3730804), both of which are essential for tolerance to low pH [42]. Genes that code for proteins associated with bile tolerance, such as choloylglycine hydrolase (MBO3729487, MBO3730178,

**Table 4. Proteins involved in the potential probiotic properties of *Lactobacillus paragasseri* UBLG-36.**

| Putative function | Protein/gene name | NCBI protein accession number |
|---|---|---|
| **pH tolerance** | Na$^+$-H$^+$ antiporter | MBO3730586 |
| | F0F1 ATP Synthase | MBO3730797—MBO3730804 |
| **Bile tolerance** | MFS transporter | MBO3729588, MBO3729710, MBO3729735, MBO3729745, MBO3729780, MBO3729802, MBO3729892, MBO3730088, MBO3730176, MBO3730177, MBO3730237, MBO3730245, MBO3730263, MBO3730265, MBO3730328, MBO3730357, MBO3730588, MBO3730672 |
| | Choloylglycine hydrolase | MBO3729487, MBO3730178, MBO3730995 |
| **Adhesion** | LPXTG cell wall anchor domain-containing protein | MBO3730086, MBO3730467, MBO3731156 |
| | Membrane lipoprotein lipid attachment site-containing protein | MBO3729742 |
| | Triose phosphate isomerase | MBO3730742 |
| | Exopolysaccharide biosynthesis protein | MBO3730885, MBO3730887 |
| | CpsD/CapB family tyrosine-protein kinase | MBO3730886 |
| | Glyceraldehyde-3-phosphate dehydrogenase | MBO3730740 |
| | MucBP domain-containing protein | MBO3731150 |
| | Elongation factor Tu | MBO3730873 |
| **Temperature** | Co-chaperone GroES | MBO3729844 |
| | Chaperonin GroEL | MBO3729843 |
| | Molecular chaperone DnaJ | MBO3729502 |
| | Molecular chaperone DnaK | MBO3729503 |
| | Cold shock protein | MBO3730847 |
| **Immunomodulation** | LTA synthase family protein | MBO3730517, MBO3730785 |
| | D-alanyl-lipoteichoic acid biosynthesis protein (*DltB*) | MBO3730324 |
| | D-alanyl-lipoteichoic acid biosynthesis protein (*DltD*) | MBO3730326 |
| | Surface exposed cell wall protein | MBO3730913 |
| | Glycosyltransferases | MBO3730551, MBO3730574 |
| | Glucosaminefructose-6-phosphate aminotransferase (*GlmS*) | MBO3729602 |
| | Peptidoglycan-binding domain-containing protein (*LysM*) | MBO3730936 |
| **Antimicrobial substances** | Bacteriocin | MBO3729785, MBO3731108 |
| | Pyruvate oxidase | MBO3730233 |
| **Defense mechanism** | Type II CRISPR-associated endonuclease Cas1 | MBO3730903 |
| | CRISPR-associated -endonuclease Cas2 | MBO3730904 |
| | Type II CRISPR RNA-guided endonuclease Cas9 | MBO3730902 |
| | Type II-A CRISPR-associated protein Csn2 | MBO3730905 |

MBO3730995), were also found in the genome. LPXTG (MBO3730086, MBO3730467, MBO3731156), a cell wall anchor domain-containing protein and membrane lipoprotein for attachment to peptidoglycan (MBO3729742), were also identified. These proteins confer bacteria the ability to interact with the surrounding environment [43]. The bioinformatic analysis also showed that UBLG-36 encode several other adhesion proteins such as mucin binding (MucBP) proteins (MBO3731150), exopolysaccharide biosynthesis protein (EPS, MBO3730885, MBO3730887), capsular polysaccharides (CPS, MBO3730886), glyceraldehyde 3-phosphate dehydrogenase (GAPDH, MBO3730740), triosephosphate isomerase (MBO3730742) and elongation factor Tu (MBO3730873). EPS and CPS can facilitate binding to biotic and abiotic surfaces [44]. Although GAPDH and triosephosphate isomerase participates in the glycolysis pathway, they can be released by the cell, facilitating the adhesion of bacteria to the host during colonization [45]. Molecular chaperones DnaK (MBO3729503), DnaJ (MBO3729502), GroES (MBO3729844), and GroEL (MBO3729843) were identified in the genome of UBLG-36. These chaperones have been studied for their ability to provide bacteria with resistance to temperature shocks [46] and to function in bacterial adhesion to the host [47]. Apart from bacteriocins that act as strong antimicrobial agents, UBLG-36 also had gene coding for pyruvate oxidase (MBO3730233). Pyruvate oxidase in bacteria plays a major role in increasing ATP production and can also provide a fitness advantage by producing hydrogen peroxide [48]. UBLG-36 harbors genes that encode for lipoteichoic acid (LTA) synthesis (MBO3730517, MBO3730785) and D-alanyl-lipoteichoic acid biosynthesis protein (MBO3730324, MBO3730326). LTA can elicit host innate immune response by interacting with membrane toll-like receptor 2, activating nuclear transcription factor-kappa B [49]. Genes such as *GlmS* that code for the enzyme glucosaminefructose-6-phosphate aminotransferase (MBO3729602) and *LysM* that code for lysing motif domain (MBO3730936) were identified in the genome. *GlmS* and *LysM* are involved in catalyzing and binding peptidoglycan, respectively, in the bacterial cell wall [50]. GTs identified by CAZyme analysis are also crucial in the formation of bacterial surface structures [51]. Surface exposed cell wall protein (MBO3730913) that is known to play an important role in bacterial interaction with the host [52] was also found in UBLG-36.

## Oxalate degradation

Inspecting the whole genome sequence of *L. paragasseri* UBLG-36 revealed the presence of both *oxc* (H5J37_003070) and *frc* (H5J37_003075) whose products (MBO3729990, MBO3729991) are putatively involved in oxalate catabolism [10]. Having shared this unique feature with other LAB such as *L. gasseri* [12] and *L. acidophilus* [10] further emphasizes the capability of *L. paragasseri* UBLG-36 to degrade oxalate. To test the ability of UBLG-36 to degrade oxalate *in vitro*, the bacteria were grown in 10 mM and 20 mM MRS-Ox and assayed at three different time points (Fig 3). UBLG-36 showed an average of 47% and 44% degradation on day 1 in 10 mM and 20 mM MRS-Ox, respectively. There was no significant difference ($p > 0.05$) in percentage oxalate degradation by UBLG-36 on day 1 between 10 mM and 20 mM MRS-Ox. When incubated in 10 mM MRS-Ox, UBLG-36 showed an average of 53% and 55% oxalate degradation on days 5 and 10, respectively. However, when incubated in 20 mM MRS-Ox, the average degradation significantly reduced to 48% on day 5 ($p < 0.01$) and 10 ($p < 0.001$) compared to that observed in 10 mM MRS-Ox on the same days. Overall, the percentage oxalate degradation in 10 mM and 20 mM MRS-Ox by UBLG-36 was significantly higher ($p < 0.001$) than the positive control, *L. acidophillus* DSM 20079, at all three time points. As expected, *L. casei* ATCC 334 that served as the negative control, showed no degradation of oxalate.

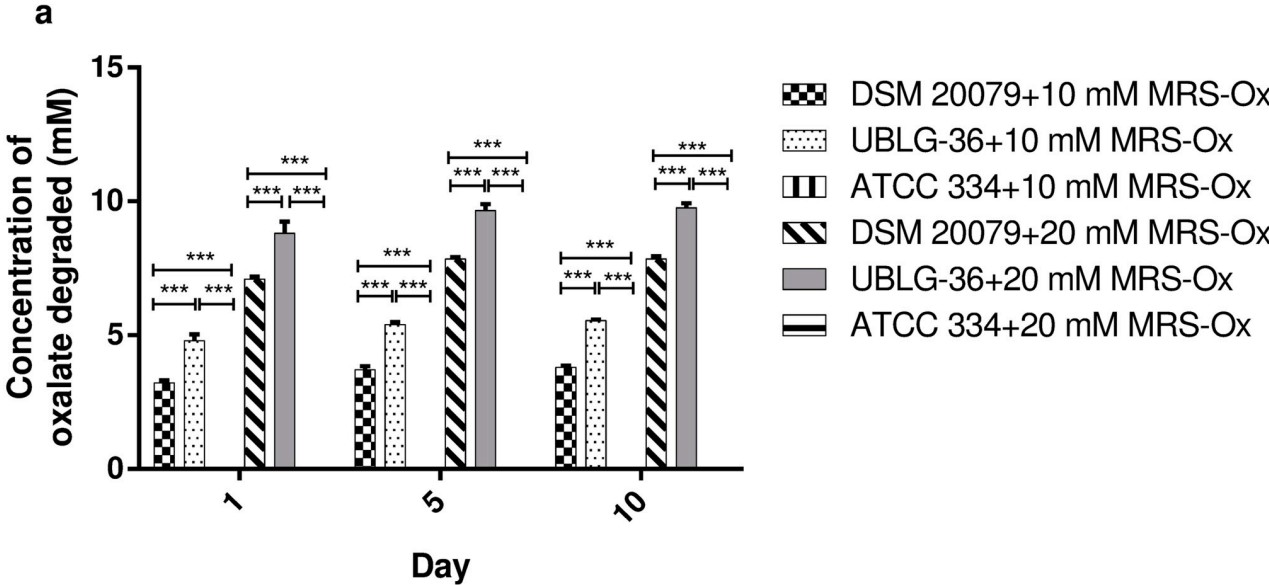

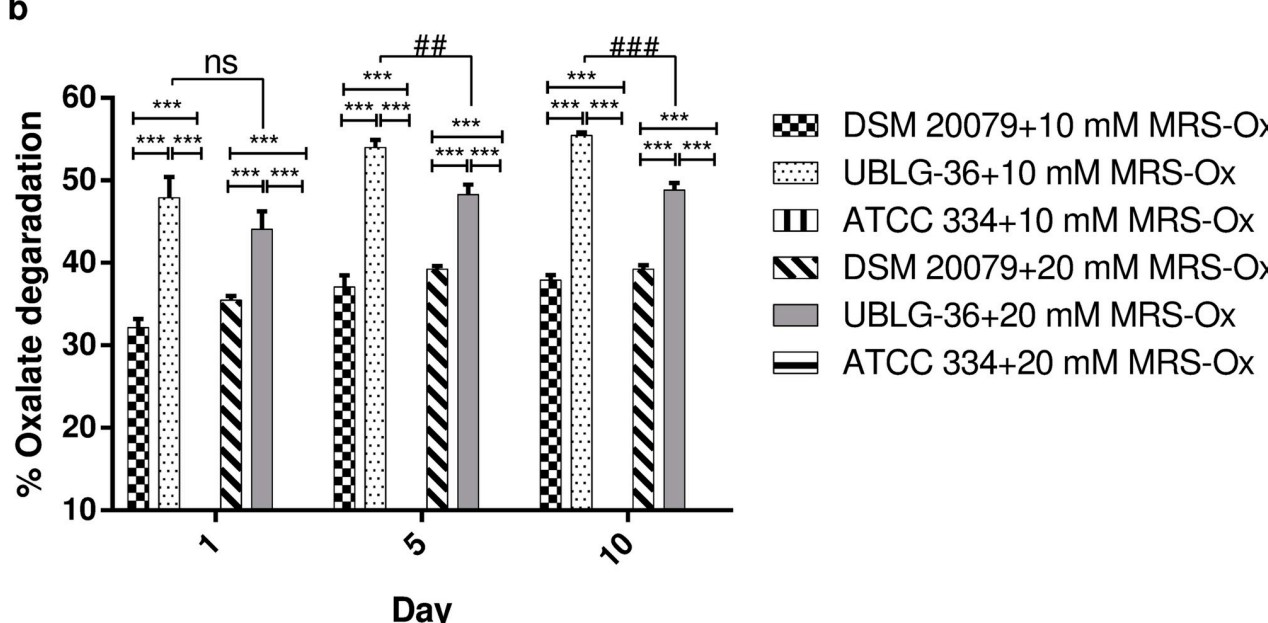

**Fig 3. Evaluation of *in vitro* oxalate degradation.** (a) The concentration of oxalate degraded by *Lactobacillus paragasseri* UBLG-36 and (b) percent oxalate degradation by *Lactobacillus paragasseri* UBLG-36 when incubated for 1, 5, and 10 days in 10 mM and 20 mM MRS-Ox. *L. acidophilus* DSM 20079 served as the positive control whereas, *L. Casei* ATCC 334 served as the negative control. *L. Casei* ATCC 334 showed no oxalate degradation. Data are presented as mean ± SEM from three experimental replicates. ***$p<0.001$, ###$p<0.001$, and ##$p<0.01$ denotes statistical significance. ns denotes non-significant.

To study the relatedness of *oxc* and *frc* found in UBLG-36 with all other representative *L. gasseri* and *L. paragasseri* strains, we performed a multiple sequence alignment and constructed a phylogenetic tree of all *oxc* and *frc* CDS. Phylogenetic analysis shows that CDS of *oxc* and *frc* in UBLG-36 is more closely related to *L. gasseri* strains HL70, EJL, and MGYG-HGUT-02387 (Fig 4), all of which have been identified as *L. paragasseri* strains based on our analysis (mentioned earlier in Fig 1 and Table 1). The *oxc* and *frc* CDS of UBLG-36 and other *L. paragasseri* strains (including *L. gasseri* strain HL70, HL75, EJL, and MGYG-HGUT-02387) together form a distinct sub-group (Fig 4), away from the three *L. gasseri* strains (HL20, ATCC33323, and DSM 14869) However, their degree of relatedness indicates a possible evolutionary inter-species exchange of oxalate catabolizing genes between *L. gasseri* and *L. paragasseri*. This observation is also supported by the result of the comparative study performed by Zhou et al. [9] that indicated a high rate of interspecies gene exchange between these sister taxa.

To the best of our knowledge, this is the first report on the oxalate degrading activity of an *L. paragasseri* strain. Given the recent taxonomic identification of *L. paragasseri*, it is not surprising that there is no research on its oxalate degrading ability. As a sister taxon of *L. gasseri*, it is expected of *L. paragasseri* strains to harbor *oxc* and *frc* and show potential oxalate metabolizing capabilities. Thus, we have observed these features not only in the genome of UBLG-36 but also in all other complete representative *L. gasseri* and *L. paragasseri* strains. When analyzed for its oxalate degrading activity *in vitro*, the results of oxalate degradation by UBLG-36 are consistent but not uniform with those reported earlier with *L. gasseri* and *L. acidophillus* strains [53]. The lack of uniformity in oxalate degradation is attributed to both species-to-species and strain-to-strain variations, as evidenced by multiple studies [12, 53, 54]. However, the highlighting feature of UBLG-36 is that the presence of *oxc* and *frc* in its genome may be responsible for its ability to degrade oxalate and with comparable oxalate degradation like some of the *L. gasseri* strains [53], UBLG-36 may prove to be an important probiotic strain.

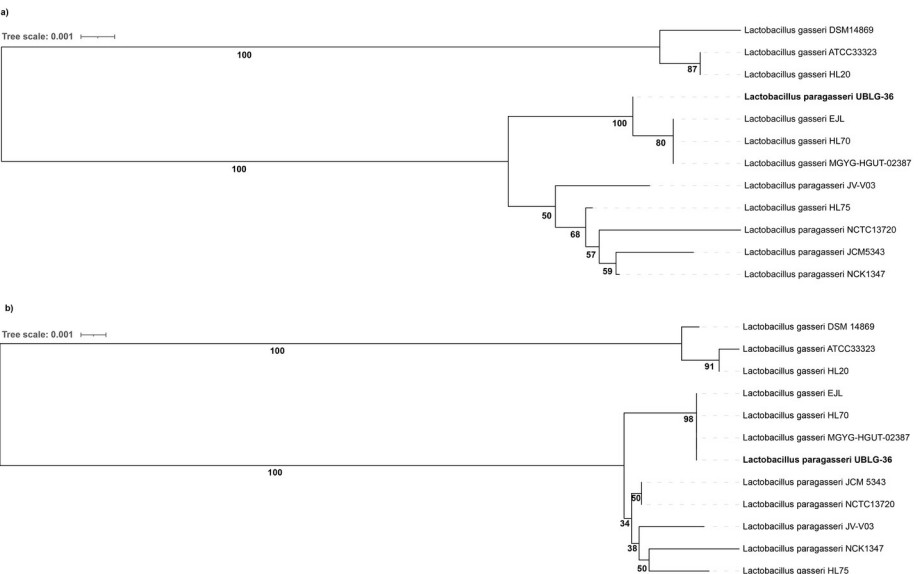

**Fig 4. Relatedness of *oxc* and *frc* of *Lactobacillus paragasseri* UBLG-36 with all other representative genomes.** (a) Phylogenetic relationship of the *oxc* coding sequence of *Lactobacillus paragasseri* UBLG-36 and other *L. gasseri* and *L. paragasseri* strains. (b) Phylogenetic relationship of the *frc* coding sequence of *Lactobacillus paragasseri* UBLG-36 and other *L. gasseri* and *L. paragasseri* strains. Phylogenetic tree construction was based on the Neighbor-Joining method with 1000 bootstrap resampling performed on MAFFT webtool. *L. paragasseri* UBLG-36 is depicted in bold.

With several potential probiotic traits discovered using bioinformatic tools, it is now essential that we also assess these traits in UBLG-36 using *in vitro* assays. Some critical *in vitro* assessments involve testing for acid and bile tolerance, cell-surface hydrophobicity, antibiotic susceptibility, immunomodulatory activity, auto-aggregation screening, production of vitamins, amines and toxins. Although our results suggest the perspective of *L. paragasseri* UBLG-36 as a new probiotic species with significant oxalate reducing capabilities, we should emphasize that further characterization of this strain in colon-simulated oxalate conditions and *in vivo* models of hyperoxaluria are necessary. Further, several other *in vivo* functional tests may be required before UBLG-36 can be classified as a probiotic strain. To this extent, we are currently performing transcriptional and functional analysis of *oxc*, *frc*, and other probiotic genes to determine whether *L. paragasseri* UBLG-36 is a probiotic strain and whether it will eventually be of use in the prophylactic treatment of renal oxalate stone.

## Conclusion

In this study, analysis of the draft genome sequence has provided evidence of the potential probiotic properties of *L. paragasseri* UBLG-36. The presence of oxalate catabolizing genes and the ability to degrade oxalate *in vitro* necessitates deeper characterization of this species which is currently underway in our laboratory. Functional profiling has illustrated genes and proteins of UBLG-36 that are most commonly shared by several important lactic acid bacteria. As a sister taxon of *L. gasseri*, an already established probiotic bacteria of the human gut microbiome with immense commercial value, *L. paragasseri* UBLG-36 may also get its due recognition but would require extensive molecular and physiological characterization. The rapid development of sequencing technologies and bioinformatic analysis has made it easier to analyze and publish genomic information of a large number of microbial species. The increasing collection of genomes in public databases will provide a reliable platform for further comparative genomic analysis that will assist in expanding our knowledge on *L. paragasseri* UBLG-36.

## Supporting information

**S1 Fig. Venn diagram distribution of CAZymes predicted by three different tools (diamond, HMMER, and hotpep) on dbCAN meta server.**
(TIF)

**S1 Table. NCBI bioproject and biosample accession numbers of representative genomes.**
(XLSX)

**S2 Table. Average nucleotide identity values of *L. paragasseri* UBLG-36 with other *L. gasseri* and *L. paragasseri* strains.**
(XLSX)

**S3 Table. Genes associated with CAZymes gene families.**
(XLSX)

**S4 Table. Bacteriocin producing secondary metabolite gene clusters.**
(XLSX)

**S5 Table. Complete prophage regions of the UBLG-36 genome.**
(XLSX)

**S6 Table. CRISPR locus of *Lactobacillus paragasseri* UBLG-36.**
(XLSX)

## Acknowledgments

The authors are thankful to Vellore Institute of Technology for providing the research amenities.

## Author Contributions

**Conceptualization:** Yogita Mehra, Pragasam Viswanathan.

**Data curation:** Pragasam Viswanathan.

**Formal analysis:** Yogita Mehra.

**Funding acquisition:** Pragasam Viswanathan.

**Investigation:** Yogita Mehra.

**Methodology:** Yogita Mehra, Pragasam Viswanathan.

**Project administration:** Pragasam Viswanathan.

**Resources:** Pragasam Viswanathan.

**Supervision:** Pragasam Viswanathan.

**Validation:** Pragasam Viswanathan.

**Writing – review & editing:** Pragasam Viswanathan.

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
