## [Decision Letter · Decision Letter 0]

16 Jul 2021

PONE-D-21-19912

High quality whole-genome sequence analysis of Lactobacillus paragasseri UBLG-36 reveals oxalate-degrading potential of the strain

PLOS ONE

Dear Dr. Viswanathan,

Thank you for submitting your manuscript to PLOS ONE. After careful consideration, we feel that it has merit but does not fully meet PLOS ONE’s publication criteria as it currently stands. Therefore, we invite you to submit a revised version of the manuscript that addresses the points raised during the review process.

Reviewer #2 had some valuable suggestions to improve the paper. Please consider them carefully.

We look forward to receiving your revised manuscript.

Kind regards,

Yanbin Yin

Academic Editor

PLOS ONE

Journal Requirements:

Reviewers' comments:

Reviewer's Responses to Questions

**Comments to the Author**

1. Is the manuscript technically sound, and do the data support the conclusions?

Reviewer #1: Yes

Reviewer #2: Yes

2. Has the statistical analysis been performed appropriately and rigorously? 

Reviewer #1: Yes

Reviewer #2: No

3. Have the authors made all data underlying the findings in their manuscript fully available?

Reviewer #1: Yes

Reviewer #2: Yes

4. Is the manuscript presented in an intelligible fashion and written in standard English?

Reviewer #1: Yes

Reviewer #2: Yes

5. Review Comments to the Author

Reviewer #1: The authors of the manuscript clearly explained the discovery and characterization of a new Lactobacillus paragasseri strain UBLG-36. The genome of the new strain was sequenced and a draft genome was assembled. The authors throughly listed genomic features and performed bioinformatic analyses including functional classification, carbohydrate-active enzyme identification, prophage identification, secondary metabolites, CRISPR-Cas, and putative probiotic properties. Wet lab experiments also verified UBLG-36’s ability to degrade oxalate, highlighting the strain’s potential in being utilized as a probiotic.

The genomic analysis conducted here are general and rather superficial, but as the authors stated, much more experiments and analysis are needed for the further verification of UBLG-36’s potential probiotic properties. Since there has yet to be a comprehensive probiotic associated genomic analysis study on the recently discovered Lactobacillus paragasseri, the results generated here may be useful to future related projects.

However, I do have some minor suggestions and questions for the authors. 1. When performing the initial phylogenomic analysis, the complete genomes of 4 L. paragasseri and 4 L. gasseri strains were used. There are currently 8 complete genomes for Lactobacillus gasseri strains on NCBI, why not use all of them for the analysis? 2. In figure3, there are too many colors for the legends and it is making it a bit difficult to differentiate between the positive negative controls and the different time points. Maybe use more labels on the graph to make the figure easier to comprehend.

Reviewer #2: This work provides a foundation to further study L. paragasseri as putative probiotic species. Genome characterization of the UBLG-36 strain through bioinformatics analyses were sufficient, although further interpretation and discussion is needed to explain the genomic results in the context of a putative probiotic strain.

Line175: Should there be any concerns that these antibiotic resistance genes are present (in accordance with EFSA regulations)? Any flanking regions that suggest possible transfer of resistance genes?

Line197: Why is carbohydrate metabolism an important trait and what might this imply towards the environmental niche whereby UBLG-36 is expected to survive? Does the CAZyme analysis imply preference for particular substrates?

Line207: What are the implications of the discovery of these bacteriocin producing clusters? Are these expected from L. gasseri / L. paragasseri?

While the title of the manuscript refers to the oxalate degrading potential of UBLG-36, there is an overall lack of emphasis in the text that describes the importance of oxalate degradation and why this is the focus of the manuscript. Perhaps, a few sentences are needed to highlight the importance of this trait and how it is valuable in probiotics and human health. In addition, it will be worth mentioning the general capability of probiotic LAB species in oxalate degradation and how this trait differs between species/strains.

Besides, additional pangenomics analyses can be carried out between the 8 selected genomes (or more if publicly available), especially in the context of the oxc and frc genes that are of main interest in this manuscript. This should provide insights as to how frequent these oxalate degradative genes are present in L. gasseri / L. paragasseri species (highly conserved?), their homology and how their presence/absence affect oxalate utilization capabilities (Turroni et al. 2007: https://sfamjournals.onlinelibrary.wiley.com/doi/10.1111/j.1365-2672.2007.03388.x). Also, this can potentially reveal differences in the putative probiotic-associated genes that were investigated in the paper, perhaps even further differentiating L. paragasseri from its sister taxa. If these are already known from previous publications, it would be worth citing and discussing.

Fig3: Statistical analysis required to show significant differences between UBLG-36 with the controls and between the two different concentrations at each time point.

Line264: It would be insightful to discuss if the percentage of oxalate degradation reported here are consistent with other probiotic L. gasseri/L. paragasseri/LAB species that were previously reported, as oxalate degradation activity is highly variable (Turroni et al. 2007, Azcarate-Peril et al. 2008). With some strains reported to have 100% oxalate degradative activity, where does UBLG-36 stand?

Line266: It appears that there are opposing trends between DSM 20079 and UBLG-36 with respect to different oxalate starting concentrations at all three timepoints. For DSM 20079, the percentage of oxalate degradation appears to be higher at 20mM compared to 10mM while for UBLG-36, the percentage of oxalate degradation appears to be lower at 20mM compared to 10mM. Was degradation at 20mM statistically different compared to 10mM for each strain? If yes, is there a proposed hypothesis as to why this difference in dose-dependent effects were observed?

Line283: Briefly expand on what are some of the important assays/phenotyping that are needed in order to establish a strain as a probiotic.

6. PLOS authors have the option to publish the peer review history of their article (what does this mean?). If published, this will include your full peer review and any attached files.

Reviewer #1: No

Reviewer #2: No

---

## [Author Response · Author response to Decision Letter 0]

30 Sep 2021

Response to reviewers

We would like to thank the reviewers for their thorough reading of the manuscript and the thoughtful comments and constructive suggestions. Acting on the revisions prompted by the reviewers’ helped in improving the quality of the manuscript. Changes in the revised manuscript has been done with track change option in MS Word.

Reviewer 1: 

Comment 1: When performing the initial phylogenomic analysis the complete genomes of 4 L. paragasseri and 4 L. gasseri strains were used. There are currently 8 genomes for L. gasseri strains on NCBI, why not use them all?

Response: We could not use the new L. gasseri strains, namely, HL20, HL70, and HL75, as they were not available at the time, we performed the analysis. Further, ANI blast analysis on strains 4M13 and MGYG-HGUT-02387 indicated identical (duplicate genomes). Therefore, we only included MGYG-HGUT-02387 in our analysis. However, we thank the reviewer for his suggestion, and in our revised manuscript, we have included these new strains and performed a new set of phylogenomic analyses (see section “Phylogenomic analysis” under “Results and Discussion”).

Comment 2. In figure3, there are too many colors for the legends and it is making it a bit difficult to differentiate between the positive negative controls and the different time points. Maybe use more labels on the graph to make the figure easier to comprehend.

Response: As per the reviewer’s suggestion, we have modified the figure to make it more legible and comprehensible (See Fig3).

Reviewer 2:

Comment 1: Should there be any concerns that these antibiotic resistance genes are present (in accordance with EFSA regulations)? Any flanking regions that suggest possible transfer of resistance genes? 

Response: We thank the reviewer for raising this concern, as it prompted us to investigate these genes more closely. We realized that our initial assessment of antimicrobial-resistant genes (AMR) using PATRIC was wrong because the output of PATRIC is based only on the homology of the query sequence with genes of those organisms that are in the PATRIC AMR gene database (which does not include any Lactobacillus or probiotic species). Moreover, we realized that the PATRIC database is not the best tool to identify putative AMR genes as PATRIC only indicates the possible AMR mechanisms and does not provide the genes that can putatively confer AMR. Therefore, we repeated the analysis by submitting the draft genome of UBLG-36 to the ResFinder tool. No, AMR genes were identified in the genome of UBLG-36. To confirm this data, we aligned the annotated sequence of UBLG-36 against the protein sequences of AMR genes in the Comprehensive Antibiotic Resistance Database (CARD). Similar to ResFinder, no resistance phenotype or gene was observed in UBLG-36 when searched in the CARD database. We have, therefore, updated this information in both the “Materials and Method” and the “Result and Discussion” sections of the revised manuscript and sincerely apologize for this mistake. 

Comment 2: Why is carbohydrate metabolism an important trait and what might this imply towards the environmental niche whereby UBLG-36 is expected to survive? Does the CAZyme analysis imply preference for particular substrates?

Response: While we were not emphasizing any particular substrate in our analysis, in our revised manuscript, we have included some brief points as to how the presence of these enzymes might be beneficial to UBLG-36 (see section “Carbohydrate-active enzymes (CAZymes)”). For example, Carbohydrate metabolism provides the main source of metabolic energy in LAB. It plays an important role in the survival and fitness of Lactobacillus in their ecological niche by contributing to cellular processes such as energy production and stress response. We believe that the presence of these enzymes will aid UBLG-36 in its survival, competitiveness, and persistence within the host.

Comment 3: What are the implications of the discovery of these bacteriocin producing clusters? Are these expected from L. gasseri / L. paragasseri?

Response: As mentioned in the “result and discussion” of the manuscript under the “Secondary Metabolites”, bacteriocins are essential antimicrobial peptides produced by LAB. Of particular interest amongst all classes of bacteriocins are the class V circular bacteriocins such as gassericin reported in our strain, L. paragasseri UBLG-36. As mentioned in the revised manuscript (line 233, “Secondary metabolites”), bacteriocins like gassericin are simple representatives of ubiquitous circular peptides with diverse physiological activities (Craik et al. 2006 Biopolymers 84:250–266). They were first identified in L. gasseri LA39 and have been shown to be active against several foodborne pathogenic bacteria (Kawai et al. 2001 Food Microbiol 18:407–415). As L. paragasseri strains are a sister taxon of L. gasseri, it is expected that L. paragasseri strains will also have bacteriocin clusters in their genome as described previously (see reference 9 in the revised manuscript). 

Comment 4: While the title of the manuscript refers to the oxalate degrading potential of UBLG-36, there is an overall lack of emphasis in the text that describes the importance of oxalate degradation and why this is the focus of the manuscript. Perhaps, a few sentences are needed to highlight the importance of this trait and how it is valuable in probiotics and human health. In addition, it will be worth mentioning the general capability of probiotic LAB species in oxalate degradation and how this trait differs between species/strains.

Response: We agree with the reviewer’s comment that the manuscript lacks an emphasis on the importance of oxalate degradation, both in general and in the context of L. paragasseri UBLG-36. Therefore, in the revised manuscript, we have included several points under “Introduction” and the “Results and Discussion” sections highlighting the relevance of discovering oxalate degrading genes in UBLG-36 and the necessity for oxalate degradation. 

Comment 5: Besides, additional pangenomics analyses can be carried out between the 8 selected genomes (or more if publicly available), especially in the context of the oxc and frc genes that are of main interest in this manuscript. This should provide insights as to how frequent these oxalate degradative genes are present in L. gasseri / L. paragasseri species (highly conserved?) If these are already known from previous publications, it would be worth citing and discussing.

Response: Although we understand that pangenomics results will be insightful, we are currently using the pangenomics data of UBLG-36 to perform several transcriptional and functional analyses on the strain in our laboratory. Therefore, we apologize for being unable to provide a comparative pangenomic analysis between UBLG-36 and other type strains, in this manuscript, at this point. However, in the “Introduction” to our revised manuscript, we have discussed few key points from a recently published pangenomic study by Zhou et al. (2020) compared the genetic diversity of 13 strains of L. gasseri and 79 strains of L. paragasseri (Line 76, see reference 9 in the revised manuscript). Further, we thank the reviewer for the suggestion and have performed analyses to determine the evolutionary conservation of oxc and frc between L. gassei and L. paragasseri species using complete genomes present in the NCBI database. We have included this new detail in the “Materials and Methods” and have discussed the results in the revised manuscript (See line 125 and 308, and Fig 4 in the revised manuscript)

Comment 6: Fig3: Statistical analysis required to show significant differences between UBLG-36 with the controls and between the two different concentrations at each time point.

Response: As per the reviewer’s suggestion, we have performed the statistical analysis and have updated the information under “Results and Discussion” in the revised manuscript (Line 294 and Fig 3)

Comment 7: It would be insightful to discuss if the percentage of oxalate degradation reported here are consistent with other probiotic L. gasseri/L. paragasseri/LAB species that were previously reported, as oxalate degradation activity is highly variable (Turroni et al. 2007, Azcarate-Peril et al. 2008). With some strains reported to have 100% oxalate degradative activity, where does UBLG-36 stand?

Response: We agree with the reviewer that some discussion on the presence of oxalate degrading genes and the results of oxalate degradation by UBLG-36 are necessary. Hence, in the revised manuscript, we have added a concise paragraph discussing the consistency of our results with other LAB strains (Line 325). We agree that the degree of oxalate degradation by LAB is highly variable and believe that it is most likely due to species- and strain-specific differences. Further, in their study, Turroni et al. (2007) showed that oxalate degradation as much as 40% was an effective trait in L. gasseri, although some strains showed 100% degradation. However, their study measured oxalate degradation over five days by inoculating the bacteria in 5 mM oxalate. Few significant differences in our study (apart from the strains used) is that UBLG-36 was subjected to higher concentrations of oxalate, and we have measured oxalate degradation for a longer duration than that reported by Turroni et al. Therefore, the notable differences in the levels of degradation between UBLG-36 and other L. gasseri or LAB species may not only be influenced by the type of strain or species but also the experimental setup. Also, to the best of our knowledge, this is the first report on the ability of an L. paragasseri strain in degrading oxalate. Therefore, based on the current results, we believe that UBLG-36 is an oxalate degrader with high potential and can show promising results when characterized further. 

Comment 8: It appears that there are opposing trends between DSM 20079 and UBLG-36 with respect to different oxalate concentrations at all three timepoints. For DSM 20079, the percentage of oxalate degradation appears to be higher at 20mM compared to 10mM while for UBLG-36, the percentage of oxalate degradation appears to be lower at 20mM compared to 10mM. Was degradation at 20mM statistically different compared to 10mM for each strain? If yes, is there a proposed hypothesis as to why this difference in dose-dependent effects were observed?

Response: The opposing trends in oxalate degradation between DSM 20079 and UBLG-36 maybe due to species-wise differences in catabolizing oxalate. Such variability has been previously reported (Turroni et al. 2007, Azcarate-Peril et al. 2008). Further, as far as dose-dependent effect is concerned, the differences in the physiological state and adaptation of the cells to the oxalate medium may also be contributing to the observed effect. We want to add that DSM 20079 was only used as a positive control for the experiment (because it harbors oxc and frc), and our analysis did not compare this strain’s effectiveness against UBLG-36. However, as per the reviewer’s suggestion, in the revised manuscript, we have shown the statistical analysis (Line 294 and Fig 3) where we have statistically analyzed the difference in oxalate degradation between UBLG-36 and DSM 20079 (positive control) on all three-time points. We have also analyzed the oxalate degradation by UBLG-36 at 10 mM and 20 mM MRS-Ox at different time points. There was no significant difference in oxalate degradation by UBLG-36 on day 1 between 10 mM and 20 mM MRS-Ox. However, when incubated in 20 mM MRS-Ox, the average degradation significantly reduced on days 5 and 10 compared to that observed in 10 mM MRS-Ox on the same days. Further, the percentage oxalate degradation in 10 mM and 20 mM MRS-Ox by UBLG-36 was significantly higher than the positive control, L. acidophillus DSM 20079, at all three-time points. 

Comment 9: Briefly expand on what are some of the important assays/phenotyping that are needed to establish a strain as a probiotic.

Response: As per the reviewer’s suggestion, we have briefly mentioned in our discussion few important functional assays that are necessary to classify a strain (or, in this case, UBLG-36) as a probiotic (Line 336)

---

## [Decision Letter · Decision Letter 1]

14 Oct 2021

PONE-D-21-19912R1High-quality whole-genome sequence analysis of Lactobacillus paragasseri UBLG-36 reveals oxalate-degrading potential of the strainPLOS ONE

Dear Dr. Viswanathan,

Thank you for submitting your manuscript to PLOS ONE. After careful consideration, we feel that it has merit but does not fully meet PLOS ONE’s publication criteria as it currently stands. Therefore, we invite you to submit a revised version of the manuscript that addresses the points raised during the review process.

Please address the reviewer #2's new comments.==============================

We look forward to receiving your revised manuscript.

Kind regards,

Yanbin Yin

Academic Editor

PLOS ONE

Journal Requirements:

Additional Editor Comments (if provided):

Reviewers' comments:

Reviewer's Responses to Questions

**Comments to the Author**

1. If the authors have adequately addressed your comments raised in a previous round of review and you feel that this manuscript is now acceptable for publication, you may indicate that here to bypass the “Comments to the Author” section, enter your conflict of interest statement in the “Confidential to Editor” section, and submit your "Accept" recommendation.

Reviewer #1: All comments have been addressed

Reviewer #2: All comments have been addressed

2. Is the manuscript technically sound, and do the data support the conclusions?

Reviewer #1: Yes

Reviewer #2: Yes

3. Has the statistical analysis been performed appropriately and rigorously? 

Reviewer #1: Yes

Reviewer #2: Yes

4. Have the authors made all data underlying the findings in their manuscript fully available?

Reviewer #1: Yes

Reviewer #2: Yes

5. Is the manuscript presented in an intelligible fashion and written in standard English?

Reviewer #1: Yes

Reviewer #2: Yes

6. Review Comments to the Author

Reviewer #1: The authors addressed my previous concerns with the included additional data and experiments regarding the phylogenetic analysis of L. gasseri strains.

Regarding concerns on Figure 3, the authors also made modifications and made the figure more comprehensible to readers.

Reviewer #2: The authors' fully addressed previous comments. I commend the time and effort that the authors dedicated to re-visit certain analyses in the manuscript, especially pertaining to the antibiotic resistance genes.

Minor comments:

line 141: It is advisable to state the factors used for the two-way ANOVA. For figure 3a, since comparisons were not made across concentrations, was a two-way ANOVA still used here or a one-way ANOVA with bacteria strain as the factor? In addition, were the statistics done excluding the negative control? Although obvious, the negative control should be included. This would allow the authors to confidently state that oxalate degradation is significantly higher in both positive control and the test strain compared to the negative control, proving that the negative control worked as expected.

line 305: Include the fact that no oxalate degradation was observed after indicating that L. casei ATCC 334 served as a negative control. This will help to avoid confusion as ATCC 334 is shown in the figure legends but is not visible in the graph itself due to lack of degradative activity.

7. PLOS authors have the option to publish the peer review history of their article (what does this mean?). If published, this will include your full peer review and any attached files.

Reviewer #1: No

Reviewer #2: No

---

## [Author Response · Author response to Decision Letter 1]

27 Oct 2021

Response to reviewers

We would again like to thank the reviewers for their thorough review of the manuscript and their constructive suggestions. Although the changes were minor, acting on the revisions prompted by the reviewers has immensely improved the manuscript. Changes in the revised manuscript have been done with the track change option in MS Word.

Reviewer 2: 

Comment 1: Line 141-It is advisable to state the factors used for the two-way ANOVA. Was a two-way or one-way ANOVA used for comparison in Fig 3a? It is advisable to include the negative control in the statistical analysis.

Response: We thank the reviewer for suggesting these changes. One-way ANOVA was used in figure 3a, and two-way ANOVA was used in figure 3b. We have stated these factors in the revised manuscript (line 140 under “Materials and methods”). Further, we have included the negative control in the statistical test, and the new figure 3 reflects these changes.

Comment 2. Line 305-Include the fact that no oxalate degradation was observed after indicating that L. casei ATCC 334 served as a negative control.

Response: As per the reviewer’s suggestion, we have included a line that L. casei 334 was used as the negative control and showed no oxalate degradation. (See line 306).

---

## [Editor Report · Decision Letter 2]

3 Nov 2021

High-quality whole-genome sequence analysis of Lactobacillus paragasseri UBLG-36

reveals oxalate-degrading potential of the strain

PONE-D-21-19912R2

Dear Dr. Viswanathan,

We’re pleased to inform you that your manuscript has been judged scientifically suitable for publication and will be formally accepted for publication once it meets all outstanding technical requirements.

Kind regards,

Yanbin Yin

Academic Editor

PLOS ONE
---

## [Editor Report · Acceptance letter]

8 Nov 2021

PONE-D-21-19912R2 

High-quality whole-genome sequence analysis of *Lactobacillus paragasseri* UBLG-36 reveals oxalate-degrading potential of the strain 

Dear Dr. Viswanathan:

I'm pleased to inform you that your manuscript has been deemed suitable for publication in PLOS ONE. Congratulations! Your manuscript is now with our production department. 

Kind regards, 

on behalf of

Dr. Yanbin Yin 

Academic Editor

PLOS ONE